# Effectiveness of art-based health education on anemia and health literacy among pregnant women in Western Nepal: A randomized controlled trial

**Hiroko Sakai** [1]*, **Rina Kawata**[2], **Rajesh Adhikari**[3], **Yoko Oda Thapa**[4], **Tulsi Ram Bhandari**[5]

1 Faculty of Nursing, Graduate School of Nursing, Kansai Medical University, Hirakata, Japan, 2 Division of Health Science, Graduate School of Medicine, Osaka University, Suita, Osaka, Japan, 3 Gynaecology Department of Pokhara Academy of Health Sciences, Pokhara, Nepal, 4 Action Research for Community Health International, Damauli, Nepal, 5 Faculty of Health Sciences, School of Health and Allied Sciences, Pokhara University, Pokhara, Nepal

* sakaihir@hirakata.kmu.ac.jp

**Data Availability Statement:** This study has the potential to identify caste class by name, educational background and area of residence, and

## Abstract

### Objective

As Nepalese pregnant women vary widely in literacy levels and cultural backgrounds and are reluctant to make decisions about their health, general interventions are insufficient to improve maternal anemia. This study aimed to assess the effectiveness of "face-to-face health education using educational material created using pictures, photos, and nomograms" in reducing anemia and improving health literacy.

### Methods

A total of 156 Nepalese pregnant women with hemoglobin (Hb) levels below 11.0 g/dl were divided into three groups: the education group received three sessions of face-to-face health education using art-based material unaffected by literacy skills; the distribution group received material used in the education group; and the control group underwent general perinatal checkups. Hb levels and health literacy scores were assessed at baseline early pregnancy (8–12 weeks) and late pregnancy (36–40 weeks). A Nepalese version of the 14-item Health Literacy Scale (HLS-14) was developed to assess health literacy.

### Results

The post-intervention three-group comparison showed a statistically significant difference (P < 0.042) in mean Hb levels after the intervention. Dunnett's test showed a statistically significant difference (P < 0.044) between the education and control groups but no significant difference between the distribution and control groups (P = 0.972). No significant differences in health literacy (total scores and subscales) were observed among the three groups before the intervention in the Kruskal-Wallis test and after the intervention (although there was a trend towards improvement). Total health literacy scores before and after the intervention were statistically significantly different for the total group and all three groups (P<0.001).

has received guidance on strict controls from the Nepal Health Research Council and the Research Ethics Committee of Morinomiya University of Medical Sciences in Japan. Therefore, this study has ethical constraints on the sharing of de-identified datasets. However, researchers who meet the criteria for access to confidential data, may send their request to share data to Kansai Medical University, to which the first author belongs. The contact details are as follows. Hiroko Sakai Email Address: sakaihir@hirakata.kmu.ac.jp.

**Funding:** • Initials of the authors who received each award: H S • Grant numbers awarded to each author: Hiroko Sakai JSPS KAKENHI Grant Numbers 18k17592,22k0026. • The full name of each funder: Grant-in-Aid for Young Scientists, Fostering Joint International Research (B) • URL of each funder website: Japan Society for the Promotion of Science (jsps.go.jp) The funders had no role in study design, data collection and analysis, decision to publish, or preparation of the manuscript./

**Competing interests:** The authors have declared that no competing interests exist.

Only the education group showed statistically significant differences in functional (P<0.012), communication (P<0.004), and critical (P<0.014) literacy subscale scores.

## Conclusion

Continuous face-to-face health education using literacy material significantly reduced anemia and improved health literacy among Nepalese pregnant women.

## Trial registration

UMIN Clinical Trials Registry (UMIN-CTR), URL: https://www.umin.ac.jp/ctr/ (Registration number: UMIN000049603).

## Introduction

Health literacy (HL) has been defined as "the cognitive and social skills that determine the motivation and ability of individuals to gain access to, understand, and use information in ways that promote and maintain good health" [1]. In developing countries, many social determinants of health affect people's lives, including poverty, gender inequality, educational disparities, exploitation, violence, and injustice. These factors contribute to illness and death among the poor and marginalized [2]. In the Federal Democratic Republic of Nepal (hereinafter referred to as Nepal), there are significant health disparities due to social determinants of health, and healthcare systems and support for health maintenance and promotion are still inadequate.

In particular, Nepalese women have difficulty making decisions and taking health-related actions of their own volition due to various social background-related factors, including religion, tradition, gender roles, and caste; thus, they need to be empowered [3]. Health literacy is a personal skill that can be developed and can promote independence in health care based on the patient's own decisions [4]. The potential of health literacy to reduce inequalities, increase health system responsiveness, and promote the achievement of the United Nations Sustainable Development Goals is gaining attention [5].

There is growing awareness worldwide of the ethical imperative for patients to participate in decision-making regarding the health care they receive. As patient decision-making has been shown to be effective in behavior change and health outcomes, there has also been recognition of the importance of improving patient health literacy and supporting patient self-determination [6, 7]. These effective health behaviors and health outcomes include reduced patient anxiety and increased patient knowledge, patient satisfaction with treatment decisions, and treatment adherence [8].

While substantial progress has been made in many aspects of healthcare delivery in Nepal, perinatal mortality rates remain high [9]. Moreover, malnutrition among pregnant women in Nepal has a variety of negative effects on mother and child.

Anemia during pregnancy is associated with an increased risk of perinatal and maternal death, preterm delivery, and low birth weight [10–13]. The lack of improvement in perinatal mortality is related to the nutritional status of pregnant women: 46% of pregnant women in Nepal have a body mass index (BMI) below 18.5, indicating a high rate of undernutrition [14]. In addition, 40% of pregnant women in Nepal are anemic, according to the latest data, with no significant improvement in the past 20 years [15].

Most of the previous studies on anemia and malnutrition among Nepalese pregnant women are cross-sectional studies. In addition, iron supplements have been distributed to all pregnant women as a national policy, but the anemia status of pregnant women before and after the intervention has not been assessed. Furthermore, there has been no evaluation of iron supplementation compared with a control group, and an association between low compliance to iron supplementation and high anemia rates among Nepalese pregnant women has been noted [16].

In view of the current lack of improvement in maternal hemoglobin (Hb) levels in Nepal, the distribution of iron supplements alone is not sufficient for reducing anemia [17]. This conclusion is suggested by the results of a study [18] of Nepalese pregnant women with low anemia prevalence, which reported higher compliance and improved Hb levels in the education plus pill count group than in the pill count alone group.

In particular, the causes of health disparities among Nepalese women are diverse, pointing to the need for long-term observation aimed at improving health literacy in addition to health outcomes when interventions for health promotion are implemented [19].

Although we could not identify any previous studies showing that health literacy level affects maternal anemia, we aimed to assess from 14 articles whether pregnant women's health literacy level is associated with pregnancy outcomes and whether effective interventions for improving pregnant women's health literacy have been established. A systematic review on this topic reported that the health literacy levels of pregnant women varied widely, and low health literacy was associated with unhealthy behaviors during pregnancy [20]. However, little is known about health literacy in Nepal. A previous study published in 2018 [21] assessed the health literacy levels of chronically ill patients and ascertained their knowledge about their disease. The results showed that 27% of respondents had adequate, 19% had low, and 54% had inadequate health literacy levels. Factors associated with inadequate health literacy included older age, female sex, low or no education, unemployment or retirement, poverty, and history of smoking or drinking. Those with adequate health literacy understood their disease or condition more significantly than those with inadequate health literacy [21].

Health literacy interventions consist of lectures, passive teaching, one-way delivery of information, distribution of brochures and leaflets, and health education sessions using visual aids [22]. Traditional methods alone are inadequate for pregnant women in Nepal, who have varying literacy levels and cultural backgrounds and are reluctant to make decisions about their own health. Health education for persons with low health literacy requires the use of materials that consider the comprehension skills of the target population, and in this context, material that includes pictures and diagrams is understood significantly better [23]. Therefore, this study aimed to evaluate the effectiveness of "face-to-face health education using educational material created with pictures, photos, and nomograms without text" to improve Hb levels in Nepalese pregnant women. Furthermore, the study sought to assess whether the intervention improved the health literacy of Nepalese pregnant women.

## Materials and methods

### Ethics statement

The survey was conducted with the approval of the Research Ethics Committee (No. 2018180) of the Kansai Medical University to which the principal investigator belongs, as well as the Institutional Review Committees of the Nepal Health Research Council (No. 358) and Western Regional Hospital (No. 64). This was an interventional study and required clinical trial enrollment prior to the start of the study. However, as we were unaware of the need for prospective clinical trial enrollment, this trial was enrolled retrospectively (UMIN Clinical Trials Registry,

UMIN000049603; enrollment date: 11/24/2022, retrospective enrollment). The authors confirm that all ongoing and related trials for this intervention are registered. The analysis in this study was conducted according to the analysis plan presented in the study protocol. The study participants were pregnant women who understood the study and provided written consent before enrollment. Nepalese research nurses, who had received prior training and understood the purpose and content of the study, administered the survey using a Nepali questionnaire with an ID number. The questions and answer choices in the questionnaire were verbally explained (Nepali) to the participants by the research nurse, and the verbal (Nepali) responses were transcribed onto the survey form. If the participants were aged <20 years, consent was obtained from both the participants and their guardians. Easily understandable explanations were provided for participants from whom informed consent was difficult to obtain.

## Study design

This study employed a randomized controlled trial (RCT) design and adhered to the Consolidated Standards of Reporting Trials (CONSORT) Statement. The study used a randomized parallel-group comparison design in which a single-blind test was used to compare [1] an education group using original educational material [2], a distribution group receiving only educational material, and [3] a control group receiving no intervention from the researchers and only a routine antenatal checkup.

## Eligibility criteria

Anemia tests were performed on 609 pregnant women with pregnancies of gestational age 8–12 weeks who consented to the study, and 201 anemic pregnant women with Hb levels of <11.0 g/dl were screened. Inclusion criteria for screening and consent were as follows: (1) pregnant women aged 16 years or older who were able to give informed consent, (2) an Hb concentration of 7.0–10.9 g/dl, (3) presence of a live single fetus in utero, (4) gestational age of 8–12 weeks at study entry, (5) no underlying disease requiring regular oral medication, (6) no cardiovascular disease, autoimmune disease, or any other condition affecting anemia, and (7) no condition that, in the opinion of the consenting health care professional, required exclusion from the study.

An obstetrician, a co-investigator at the collaborating institution, verified the eligibility of the participants. Five research nurses explained the study to women diagnosed with gestational anemia and received a signed research consent form.

## Randomization and allocation

The participants were assigned to one of three groups by block randomization according to a six-block, computer-generated randomization order created by a statistician not involved in the study. Research nurses then administered the surveys and interventions according to the randomization list. The five research nurses who administered the surveys and interventions were aware of the group allocations but were blinded to all participants and to the analysts of the survey results. The data were statistically analyzed using IBM SPSS Statistics, version 28.0 (IBM Japan, Tokyo, Japan) (Fig 1).

## Study area and participants

Participants were recruited from pregnant Nepalese women who came to Western Regional Hospital for prenatal care. The Western Regional Hospital is the only public general hospital in Pokhara, the second largest city in Nepal, located approximately 200 km west of the capital

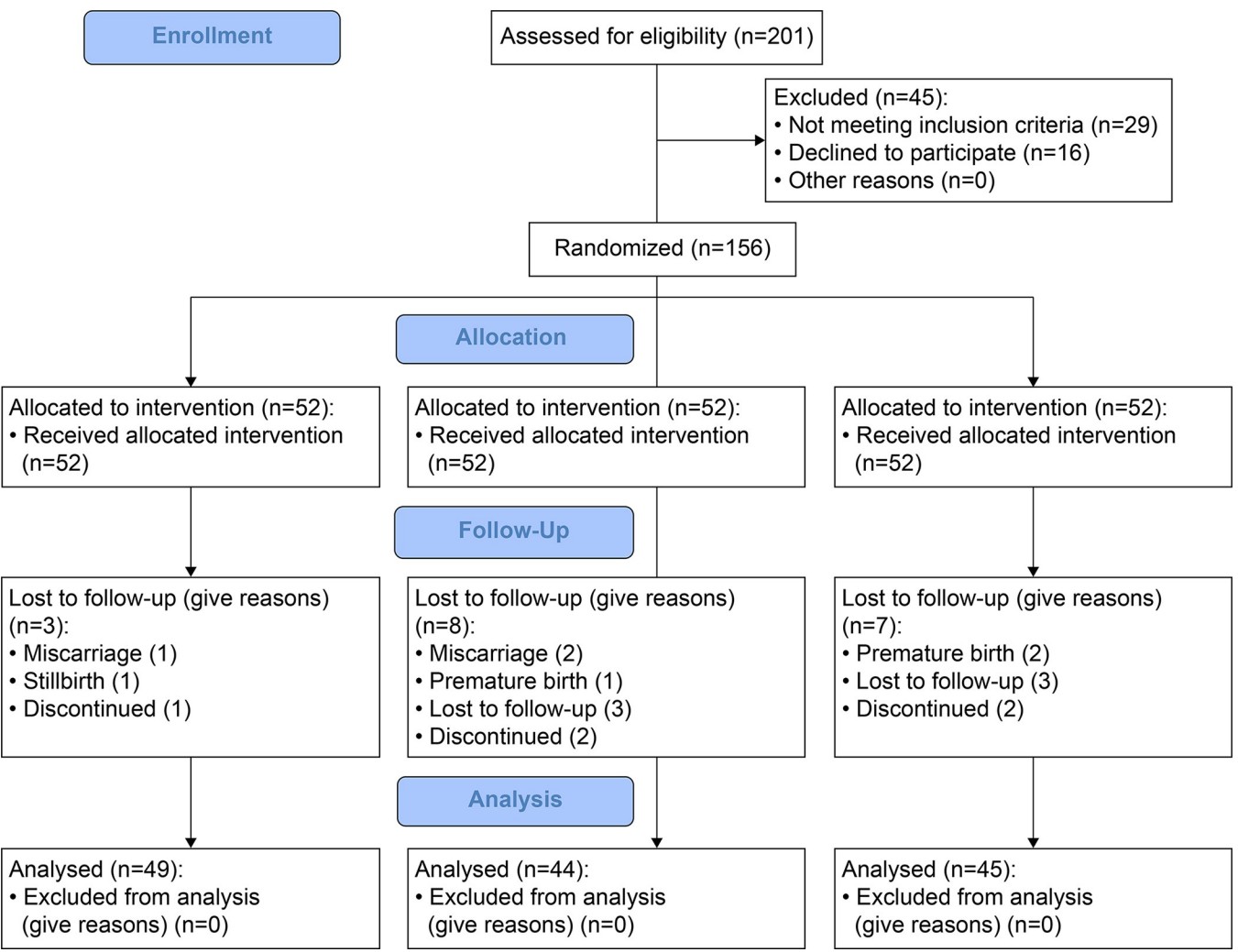

**Fig 1. Flowchart of study participation and follow-up at Western Regional Hospital.**

Kathmandu. Furthermore, it has the highest number of deliveries among the facilities in Pokhara. Pregnant women from surrounding areas (urban, mountain, and rural) use this hospital, and it was chosen as a suitable location to study pregnant women with social determinants of health that contribute to health disparities. The date range for participant recruitment and follow-up survey was from March 2019 to March 2020.

## Baseline and follow-up surveys

All participants completed a baseline survey (questionnaire including Hb levels, socioeconomic status, health status, health literacy scale, and height and weight measurements) at the first parental checkup. A follow-up survey (including Hb levels, confirmation of iron medication, and health literacy scale) was conducted at 36–40 weeks of gestation. Data were collected by research nurses through face-to-face interviews using the questionnaire. Five research nurses who had received prior training from the principal investigator and co-investigators on how to conduct interviews using the questionnaire administered the survey. Hb measurements were performed by laboratory technicians at collaborating facilities using the same equipment and analytical methods.

## Intervention

The education group underwent three individual health education sessions during all gestational periods. Health education was provided by Nepalese research nurses trained by the principal investigator and co-investigators. Participants received three health education sessions: at the 8–12 weeks baseline survey, 20–24 weeks prenatal checkup, and 30–34 weeks prenatal checkup. The health education lasted approximately 10 minutes per session and was conducted face-to-face. Original text-free material consisting of pictures, photographs, and nomograms was used in the health education sessions. Pregnant women in the distribution group received only the original educational material and did not receive individualized health education. Pregnant women in the control group received a general prenatal health examination; however, they received the original educational material in the third trimester (36–40 weeks) to ensure that they were not disadvantaged.

We also developed a teaching manual for the research nurses who conducted the education sessions. The manual described perinatology, nutrition, lifestyle, and recipes. The opinions of two Nepalese obstetricians were also incorporated to determine whether the text matched the description of the original educational material and whether the content was appropriate for education targeting Nepalese pregnant women.

The face-to-face intervention corresponded to the level of understanding, amount of knowledge, and health concerns of pregnant women and included the following items: medical knowledge about anemia among pregnant women, the effects of anemia on mother and child, the importance of iron supplementation, Nepali women's food culture and nutritional imbalances, junk food consumption, iron-rich food consumption and cooking methods to prevent and improve anemia, and precautions and side effects when taking iron pills and how to deal with them. As part of the nutrition education program, we used illustrations and photos to introduce menus and recipes that improve anemia using inexpensive and locally available food items.

## Data collection

**Baseline study.** Information on social background relating to health determinants was obtained in a baseline study.

Questions included family name to identify ethnicity and caste; age (date of birth); gestational age; place of residence; religion; age at marriage; family structure; pregnancy history; number of children; birth interval; employment of mother and husband; occupation of mother, husband, and father; household income; education of mother and husband; and literacy of mother and husband. Questions on health status included underlying medical conditions and pregnancy complications. Questions regarding daily life included burden (working hours and sense of burden); husband's interest in and understanding of prenatal care; smoking and drinking; exposure to secondhand smoke; frequency of meals; frequency of meat, fish, and legume consumption; age; treatment history of various infectious diseases, including HIV and AIDS; and iron supplementation. The questionnaire was developed by the principal investigator with reference to previous studies and was translated into Nepali.

**Intervention follow-up study.** A follow-up study was conducted in all groups. Questionnaires were administered during prenatal checkups at a gestational age of 36–40 weeks. The questionnaire consisted of the 14-item Health Literacy Scale (HLS-14) and iron medication status. Hb levels were also tested.

## Primary endpoint

**Hemoglobin level (g/dl).** Blood samples were collected by research nurses and analyzed at laboratories in cooperating research facilities for the assessment of blood Hb levels. An Hb

cutoff value (according to the World Health Organization [WHO]) of <11.0 g/dl was defined as anemia in pregnant women. Pregnant women with anemia were screened by conducting a baseline study of pregnant women with pregnancies of gestational age 8–12 weeks. Women diagnosed as anemic during this screening were recruited for the intervention study.

Hb levels were assessed again at 36–40 weeks of gestational age as a follow-up study. The primary outcome measure was the change in Hb levels between baseline and follow-up studies in the three groups.

## Secondary endpoints

**Health literacy.** The HLS-14 developed by Suka et al. [24] was used as a measure of health literacy in Japanese patients. The HLS-14 is a comprehensive health literacy scale consisting of 14 items (5 functional literacy items on a 25-point scale, 5 communicative literacy items on a 25-point scale, and 4 critical literacy items on a 20-point scale), with a 70-point scale. Functional literacy refers to the basic skills of reading and writing to function effectively in daily life. Communicative literacy refers to more advanced skills for actively participating in daily life, obtaining information, understanding health status from various forms of communication and applying new information to changing health conditions. Critical literacy refers to more advanced skills in critically analyzing information and using that information to better health conditions.

The HLS-14 scale has demonstrated reliability and validity primarily with Japanese adults and has been validated in other countries [25].

## Development of a Nepalese version of the HLS-14

Approval for the development of the Nepalese version of the HLS-14 was obtained from the corresponding author in Japan, the developer of the HLS-14 [24]. The equivalence of interpretations of the meaning of each item was discussed among the two Nepalese research collaborators, the Japanese principal investigator, and a Japanese researcher living in Nepal. The constructs of the original HLS-14 and their appropriateness for application among Nepalese people were discussed. Discussions were conducted in English and Nepali.

To ensure equivalence of meaning and interpretation, the original questionnaire was translated directly from Japanese into Nepali. This was done by one Nepali with a master's degree in economics, who is a native speaker of Nepali and fluent in English and Japanese, and one Japanese national with a master's degree in public health living in Nepal. Translations were made from the Nepali translation into English and from English into Japanese by translators (one native Nepali speaker fluent in English and one native Japanese speaker fluent in English) who were unaware of the purpose of this study.

It was then confirmed that the four researchers had a common understanding of each item and of the words in each item and that equivalence of meanings and items could be guaranteed. As a result, one word was modified for one item to make it more understandable. After the abovementioned process, the Nepali version of the HLS-14 questionnaire was completed.

Five Nepalese research nurses who understood the main purpose of the study were asked to check the Nepali version of the HLS-14 for any inconsistencies or discrepancies in meaning or interpretation.

## Body mass index

Height and weight measurements were taken during the first prenatal checkup (baseline study). The seca-213 portable height meter was used for height measurement, and the Tanita BC-314 (Tanita Corp., Tokyo, Japan) body composition scale was used for weight

measurement. Weights were compared with three other scales to ensure that there were no errors. BMI was calculated as weight (kg)/square of height ($m^2$). The WHO cutoff values for BMI were used: underweight, <18.5; normal weight, 18.5–24.9; and overweight/obese, >25.0.

## Supplementation of Hemferon-S tablets

All participants in the intervention and control arms received one tablet of Fefo (200 mg of dried Ferrous Sulphate + 0.40 mg Folic Acid) supplement once daily. The Government of Nepal launched the Iron Intensification Program (IIP) in 2003 [26], under which iron and folic acid are distributed to all pregnant women. The pregnant women are provided supplements from the first perinatal checkup until 45 days postpartum.

## Sample size

In a previous study, 320 pregnant Nepalese women were randomly divided into four groups (education group, iron count group, education + iron count group, and control group), and the effects of health education were examined in terms of Hb levels and anemia incidence [18]. To demonstrate the effect of health education using pictures and graphs, we estimated and calculated an effect size of 0.3571, α = 0.05, power = 0.95, and a dropout rate of 10%, with a total sample size of 138 cases and 46 cases each in the three groups (Gpower v.3.1).

## Analysis method

Using confirmatory factor analysis, fit indices (comparative fit index, Tucker-Lewis index, and root mean square error of approximation) were calculated for 609 pregnant women to confirm the factor structure of the model of the Nepali version of the HLS scale and validate the goodness of fit. Furthermore, Cronbach's Alpha coefficient was used to evaluate the reliability of the scale.

Descriptive statistics were used to analyze the data and calculate frequencies and percentages to describe the characteristics of the study participants. Continuous variables are summarized using the mean and standard deviation (SD). Fisher's exact test and the Chi-square test were used to analyze the associations of categorical variables among the three groups. Analysis of variance (ANOVA) was used to evaluate the primary outcome (Hb levels). To assess health literacy (secondary outcome), the Kruskal Wallis test was used to compare between the three groups on the health literacy scale, and the Wilcoxon signed rank sum test was used to calculate the change in total health literacy scale scores and subscale scores before and after the intervention. All p-values less than 0.05 were considered statistically significant. Statistical analysts analyzed data blinded to grouping. All outcomes were analyzed using all available data in an intention-to-treat framework. The data were statistically analyzed using IBM SPSS Statistics, version 28.0 (IBM Japan, Tokyo, Japan).

## Results

Out of the 609 pregnant women with pregnancies of gestational age 8–12 weeks who underwent prenatal health examinations during the study period and who consented to the study for anemia screening, 201 were diagnosed as anemic pregnant women. The following patients were excluded: 16 women who did not consent to the study, 1 woman with heart disease, 2 women with twins, 1 woman with severe anemia, and 25 pregnant women over 13 weeks of gestation. A total of 156 pregnant women participated in the study and were randomly assigned to one of the three groups, each with 52 pregnant women. Eighteen participants withdrew from the study follow-up due to miscarriage, premature birth, stillbirth, or transfer to

another hospital, leaving a total of 138 participants in the final analysis: 49 in the education group, 44 in the distribution group, and 45 in the control group (Table 1). The participants had a mean age of 24.4±4.5 years and a mean age at marriage of 20.2±3.2 years; 23 (16.7%) were in their teens at marriage, and 72 (52.2%) were in their 20s. The age at first childbirth was 21.1±3.2 years. One hundred and twelve (81.2%) participants were urban dwellers, 26 (18.8%) were rural dwellers, 127 (92.0%) were Hindu, 68 (49.3%) were extended family members, and 70 (50.7%) were immediate members. By ethnicity and caste, 65 (47.1%) were Brahman Chettri, a high caste; 40 (29.0%) were Janajati; 29 (21.0%) were Dalit; and 4 (2.9%) were Muslim, Madeshi, and Others. Furthermore, 77 (55.8%) had first pregnancies, 8 (18.1%) had a previous delivery 1–2 years ago, and 53 (38.4%) had a previous delivery more than 3 years ago; only one woman had a history of Worms (parasite infection). None of the participants smoked; one quit smoking during the pregnancy. Thirty-four (24.6%) women had family members who smoked, 112 (88.4%) were taking iron tablets, 134 (97.1%) of the participants and 130 (94.2%) of their husbands were literate, 3 (2.2%) of the participants and 5 (3.6%) of their husbands had no primary education, and 113 (81.9%) of the participants were housewives. 'Migrant worker' was the most common occupation status of the husbands of the participants (58 [42.0%]), and 97 (70.3%) of the households had a monthly household income of Rs 10,000–49,999 (Table 1).

No significant differences in pre-intervention Hb values were observed at baseline among the three groups (F(2,135), 0.218; P = 0.804). The Hb levels before and after the intervention were 10.20±0.62 g/dl and 11.58±0.65 g/dl in the education group (t(48) = 11.303; P<0.001), 10.19±0.69 g/dl and 11.15±1.08 g/dl in the material distribution group (t(43) = 5.286; P<0.001), and 10.27±0.51 g/dl and 11.11±1.20 g/dl in the control group (t(44) = 4.983; P<0.001), respectively. The results of the post-intervention three-group comparison showed a statistically significant difference (F(2,135), 3.253; P < 0.042) in mean Hb levels after the intervention. Dunnett's test showed a statistically significant difference (P<0.044) between the education and control groups and no significant difference between the distribution and control groups (P = 0.972) (Table 2).

The results of the confirmatory factor analysis on the health literacy scale applied to the scale developer's model showed reasonable goodness of fit ($\chi2$ = 347.090(64), p < .0001) and Kaiser–Meyer–Olkin measure of sampling adequacy (KMO = 0.903) results. The factor loadings of all items were 0.614–0.962, and all Cronbach's alpha coefficients for the subscales were above 0.970.

No significant differences in health literacy (total scores and subscales) were observed among the three groups before the intervention in the Kruskal-Wallis test. Health literacy scores (total score and subscales) among the three groups after the intervention were not significantly different, although there was a trend toward improvement (Table 3). In comparing changes in the health literacy scale (total and subscale scores) before and after the intervention, a statistically significant difference in total health literacy scores was observed for the total group and all three groups (P<0.001).

The median (interquartile range) pre- and post-intervention total scores were 59.5 (53.8–70.0) and 100.50 (92.8–120.0) in the total group, 61.0 (54.52–69.75) and 108.50 (94.0–118.0) in the education group, 56.0 (51.25–70.0) and 96.0 (77.25–117.5) in the distribution group, and 58.50 (55.0–70.0) and 98.5 (96.0–120.0) in the control group, respectively. Only the education group showed statistically significant differences in functional literacy, communicative literacy, and critical literacy subscale scores. The median (interquartile range) pre- and post-intervention scores for functional literacy were 25.0 (19.25–25.0) and 25.0 (20.0–25.2) in the education group (P<0.012, a statistically significant difference), 22.5 (18.25–25.0) and 21.0 (18.25–25.0) in the distribution group (P = 0.892), and 20.50 (19.0–25.0), and 21.0 (19.0–25.0) in the control group (P = 0.127, a statistically insignificant difference), respectively. The

**Table 1. Characteristics of the participants (n = 138).**

| | | Total (n = 138) | Education group (n = 49) | Distribution group (n = 44) | Control Group (n = 45) |
|---|---|---|---|---|---|
| **Age (years)** | Mean±SD (range) | 24.4 ± 4.5 (16–38) | 24.6 ± 4.1 (16–33) | 25. 2 ±4.7 (17–36) | 23.5± 4.7 (16–38) |
| **Age of marriage (years)** | Mean±SD (range) | 20.0±3.4 (14–31) | 20.1 ± 3.0 (15–28) | 20.1 ± 3.8 (14–31) | 19.9 ± 3.5 (14–28) |
| **Age of first childbirth (years)** | Mean±SD (range) | 21.1±3.2 (15–32) | 20.8 ± 3.0 (16–27) | 21.0 ± 2.7 (17–26) | 21.6 ± 3.8 (15–32) |
| | | n (%) | | | |
| **Residence, n (%)** | | | | | |
| | Urban | 112 (81.2) | 38(77.6) | 36(81.8) | 38(84.4) |
| | Rural | 26 (18.8) | 11(22.4) | 8(18.2) | 7(15.6) |
| **Religion, n (%)** | | | | | |
| | Hindus | 127 (92.0) | 44 (89.8) | 41 (93.2) | 41 (93.3) |
| | Buddhist | 6 (4.3) | 3 (6.1) | 1 (2.3) | 2 (4.4) |
| | Muslim | 0 (0.0) | 0 (0.0) | 0 (0.0) | 0 (0.0) |
| | Others | 5 (3.6) | 2 (4.1) | 2 (4.5) | 1 (2.2) |
| **Type of family, n (%)** | | | | | |
| | Extended family | 68 (49.3) | 27(55.1) | 18(40.9) | 23(51.1) |
| | Immediate family | 70 (50.7) | 22(44.9) | 26(59.1) | 22(48.9) |
| **Ethnicity, caste, n (%)** | | | | | |
| | Bramin, Chettri | 65 (47.1) | 23(46.9) | 17(38.6) | 25(55.6) |
| | Janajati | 40 (29.0) | 15(30.6) | 13(29.5) | 12(26.7) |
| | Dalit | 29 (21.0) | 11(22.4) | 12(27.3) | 6(13.3) |
| | Muslim, Madeshi, Others | 4 (2.9) | 0 (0.0) | 2 (4.5) | 2 (4.4) |
| **Duration from last delivery, n (%)** | | | | | |
| | First pregnancy | 77 (55.8) | 21(42.9) | 24(54.5) | 32(71.1) |
| | <1 year | 0 (0.0) | 0 (0.0) | 0 (0.0) | 0 (0.0) |
| | 1–2 years | 8 (18.1) | 4(8.2) | 3(6.8) | 1(2.2) |
| | 3 or more years | 53(38.4) | 24(49.0) | 17(38.6) | 12(26.7) |
| **Treatment history, n (%)** | | | | | |
| | None | 137 (99.3) | 49(35.8) | 43(31.4) | 45(32.8) |
| | Worms (parasite infection) | 1(0.7) | 0(0.0) | 1(100.0) | 0(0.0) |
| **Smoking history, n (%)** | | | | | |
| | No, never | 137 (99.3) | 49(100.0) | 43(97.7) | 45(100.0) |
| | Stopped during pregnancy | 1 (0.7) | 0(0.0) | 1(2.3) | 0(0.0) |
| **Family member's smoking history, n (%)** | | | | | |
| | Yes | 34 (24.6) | 13(26.5) | 10(22.7) | 11(24.4) |
| | No | 104 (75.4) | 36(73.5) | 34(77.3) | 34(75.6) |
| **History of alcohol drinking, n (%)** | | | | | |
| | Yes | 0 (0.0) | 0(0.0) | 0(0.0) | 0(0.0) |
| | No | 138 (100.0) | 49(35.5) | 44(31.9) | 45(32.6) |
| **Taking iron tablets, n (%)** | | | | | |
| | Yes | 122 (88.4) | 44(89.8) | 39(88.6) | 39(86.7) |
| | No | 16 (11.6) | 5(10.2) | 5(11.4) | 6(13.3) |
| **Literate, n (%)** | | | | | |
| | Yes | 134(97.1) | 49(100.0) | 43(97.7) | 42(93.3) |
| | No | 4 (2.9) | 0(0.0) | 1(2.3) | 3(6.7) |

*(Continued)*

**Table 1.** (Continued)

| | | Total (n = 138) | Education group (n = 49) | Distribution group (n = 44) | Control Group (n = 45) |
|---|---|---|---|---|---|
| **Literate husband, n (%)** | | | | | |
| | Yes | 130 (94.2) | 47(95.9) | 41(93.2) | 42(93.3) |
| | No | 8 (5.8) | 2(4.1) | 3(6.8) | 3(6.7) |
| **Education level (Wife), n (%)** | | | | | |
| | Non-educated | 3 (2.2) | 1 (2.0) | 1 (2.3) | 1 (2.2) |
| | Primary | 13 (9.4) | 6 (12.2) | 3 (6.8) | 4 (8.9) |
| | Secondary | 50 (36.2) | 15 (30.6) | 19 (43.2) | 16 (35.6) |
| | Higher Secondary | 41 (29.7) | 15 (30.6) | 11 (25.0) | 15 (33.3) |
| | Bachelor | 26 (18.8) | 10 (20.4) | 8 (18.2) | 8 (17.8) |
| | Higher than Bachelor | 5 (3.6) | 2 (4.1) | 2 (4.5) | 1 (2.2) |
| **Education level (Husband), n (%)** | | | | | |
| | Non-educated | 5 (3.6) | 1 (2.0) | 3(6.8) | 1(2.2) |
| | Primary | 20 (14.5) | 5(10.2) | 9(20.5) | 6(13.3) |
| | Secondary | 42(30.4) | 16(32.7) | 9(20.5) | 17(37.8) |
| | Higher Secondary | 39 (28.3) | 15(30.6) | 13(29.5) | 11(24.4) |
| | Bachelor | 28 (20.3) | 11(22.4) | 9(20.5) | 8(17.8) |
| | Higher than Bachelor | 4 (2.9) | 1(2.0) | 1(2.3) | 2(4.4) |
| **Occupation (Wife), n (%)** | | | | | |
| | Government job | 2(1.4) | 2(4.1) | 0(0.0) | 0(0.0) |
| | Agriculture | 5 (3.6) | 2(4.1) | 2(4.5) | 1(2.2) |
| | Labor | 1 (0.7) | 0(0.0) | 1(2.3) | 0(0.0) |
| | Business | 6(4.3) | 2(4.1) | 2(4.5) | 2(4.4) |
| | Housewife | 113 (81.9) | 38(77.6) | 36(81.8) | 39(86.7) |
| | Others | 11(8.0) | 5(10.2) | 3(6.8) | 3(6.7) |
| **Occupation (Husband), n (%)** | | | | | |
| | Government job | 6(4.3) | 2(4.1) | 0(0.0) | 4(8.9) |
| | Agriculture | 5 (3.6) | 1(2.0) | 0(0.0) | 4(8.9) |
| | Labor | 15 (10.9) | 6(12.2) | 5(11.4) | 4(8.9) |
| | Business | 22 (15.9) | 6(12.2) | 8(18.2) | 8(17.8) |
| | Migrant workers | 58 (42.0) | 25(51.0) | 20(45.5) | 13(28.9) |
| | Unemployed | 6 (4.3) | 1(2.0) | 1(2.3) | 4(8.9) |
| | Others | 26 (18.8) | 8(16.3) | 10(22.7) | 8(17.8) |
| **Household income, n (%)** | | | | | |
| | Below 9,999Rs | 3 (2.2) | 1(2.0) | 2(4.5) | 0(0.0) |
| | 10,000–19,999Rs | 18 (13.0) | 7(14.3) | 4(9.1) | 7(15.6) |
| | 20,000–29,999Rs | 37 (26.8) | 11(22.4) | 13(29.5) | 13(28.9) |
| | 30,000–39,999Rs | 23 (16.7) | 7(14.3) | 6(14.3) | 10(22.2) |
| | 40,000–49,999Rs | 19 (13.8) | 6(12.2) | 7(15.9) | 6(13.3) |
| | 50,000–59,999Rs | 11 (8.0) | 6(12.2) | 2(4.5) | 3(6.7) |
| | 60,000–69,999Rs | 4(2.9) | 2(4.1) | 1(2.3) | 1(2.2) |
| | 70,000–79,999Rs | 4 (2.9) | 2(4.1) | 1(2.3) | 1(2.2) |
| | 80,000–89,999Rs | 2 (1.4) | 2(4.1) | 0(0.0) | 0(0.0) |
| | 90,000–99,999Rs | 2 (1.4) | 1(2.0) | 1(2.3) | 0(0.0) |
| | 100,000Rs or more | 15 (10.9) | 4(8.2) | 7(15.9) | 4(8.9) |

[a]1 Nepalese Rupee = 0.008269 United States Dollars (27[th] June 2020)

SD, standard deviation

**Table 2. Comparison of primary outcome variables at baseline and endline in the three groups (n = 138).**

| | | Education group (n = 49) | Distribution group (n = 44) | Control Group (n = 45) | P[a] | P[b] |
|---|---|---|---|---|---|---|
| | | Mean±SD | Mean±SD | Mean±SD | | |
| **Hb level g/dl** | Baseline | 10.20±0.62 | 10.19±0.69 | 10.27±0.51 | 0.804 | |
| | Endline | 11.58±0.65 | 11.15±1.08 | 11.11±1.20 | <0.042* | |
| | | Education Group vs Control Group | | | | <0.044* |
| | | Distribution Group vs Control Group | | | | 0.972 |

[a] Analysis of variance (ANOVA) $P<0.05$*, $P<0.001$**
[b] Dunnett's test $P<0.05$*, $P<0.001$**
SD, standard deviation
Statistical significance was set at $p<0.05$.

median (interquartile range) communication literacy scores before and after the intervention were 20.5 (20.0–25.0) and 22.0 (20.0–25.0) in the education group (P<0.004, a statistically significant difference), 20.5 (20.0–25.0) and 20.5 (20.0–25.0) in the distribution group (P = 0.527), and 22.0 (20.0–25.0) and 21.5 (20.0–25.0) in the control group (P = 1.000), respectively. The median (interquartile range) critical literacy scores before and after the intervention were 17.0 (16.0–20.0) and 17.5 (16.5–20.0) in the education group (P<0.014, a statistically significant difference), 16.5 (16.0–20.0) and 16.5 (16.0–20.0) in the distribution group

**Table 3. Change in health literacy scale scores (total score and median subscale score [interquartile range]) before and after intervention (n = 138).**

| | | Baseline | Endline | Z-score | P[b] |
|---|---|---|---|---|---|
| **Health literacy total score** | Total Group (n = 138) | 59.5(53.8–70.0) | 100.50(92.8–120.0) | 8368.000 | <0.001** |
| | Education Group (n = 49) | 61.0 (54.52–69.75) | 108.50(94.0–118.0) | 1128.000 | <0.001** |
| | Distribution Group (n = 44) | 56.0(51.25–70.0) | 96.0(77.25–117.5) | 820.000 | <0.001** |
| | Control Group (n = 45) | 58.5(55.0–70.0) | 98.5(96.0–120.0) | 990.000 | <0.001** |
| | **P[a]** | 0.580 | 0.053 | | |
| **Functional literacy score** | Total Group (n = 138) | 23.5(19.0–25.0) | 22.0(19.0–25.0) | 418.500 | <0.013* |
| | Education Group (n = 49) | 25.0(19.25–25.0) | 25.0(20.0–25.0) | 36.000 | <0.012* |
| | Distribution Group (n = 44) | 22.5(18.25–25.0) | 21.0(18.25–25.0) | 31.500 | 0.892 |
| | Control Group (n = 45) | 20.5(19.0–25.0) | 21.0(19.0–25.0) | 76.500 | 0.127 |
| | P[a] | 0.481 | 0.356 | | |
| **Communicative literacy score** | Total Group (n = 138) | 21.5(20.0–25.0) | 22.0(20.0–25.0) | 261.000 | <0.025* |
| | Education Group (n = 49) | 20.5(20.0–25.0) | 22.0(20.0–25.0) | 75.000 | <0.004* |
| | Distribution Group (n = 44) | 20.5(20.0–25.0) | 20.5(20.0–25.0) | 10.500 | 0.527 |
| | Control Group (n = 45) | 22.0(20.0–25.0) | 21.5(20.0–25.0) | 14.000 | 1.000 |
| | P[a] | 0.755 | 0.223 | | |
| **Critical literacy score** | Total Group (n = 138) | 17.0(16.0–20.0) | 17.0(16.0–20.0) | 41.500 | 0.131 |
| | Education Group (n = 49) | 17.0(16.0–20.0) | 17.0(16.0–20.0) | 28.000 | <0.014* |
| | Distribution Group (n = 44) | 16.5(16.0–20.0) | 16.5(16.0–20.0) | 0.000 | 0.317 |
| | Control Group (n = 45) | 17.0(16.0–20.0) | 16.5(16.0–20.0) | 0.000 | 0.157 |
| | P[a] | 0.681 | 0.356 | | |

[a] Kruskal-Wallis test $P<0.05$*, $P<0.001$**
[b] Wilcoxon signed rank test $P<0.05$*, $P<0.001$**
Statistical significance was set at $p<0.05$.

(P = 0.317), and 17.0 (16.0–20.0) and 16.5 (16.0–20.0) in the control group (P = 0.157), respectively.

## Discussion

Iron deficiency anemia among Nepalese women is a serious public health problem that has been addressed but has yet to be resolved. The mean age of first childbirth of the women in this study was as low as 21.1 ± 3.2 years and more than half were in their teens, suggesting that anemia had persisted since adolescence. The study showed that Hb levels and health literacy can be improved by creating text-free material and continuing face-to-face individualized health education from early pregnancy. The health education group showed significantly improved Hb levels compared with the handout and control groups.

One art-based reproductive health intervention study in India [27] used a health literacy improvement approach to strengthen women's decision-making power regarding contraceptive knowledge, marital communication, family planning decision-making, and women's reproductive rights through street theater and puppet shows. Lori et al. [28] also conducted a study in Ghana aimed at improving health literacy to increase the ability to understand health messages and practice healthy behaviors. They used a "take action" strategy using demonstrations, role-plays, and puppet shows to help women understand and practice health messages. The results showed that health literacy improved in the intervention group compared with that in the group that received general maternity care. In addition, one prospective cohort study in Ghana [29] examined the effects of storytelling, peer support, demonstrations, and teach-back on improving health literacy. Prenatal and postpartum women in Ghana who participated in the study showed an improved understanding of health messages and improved health literacy. Therefore, health literacy interventions targeting women in developing countries are likely to contribute to improved reproductive health.

The literacy rate of the participants in this study was more than 90% for both couples, which is comparable with the literacy rate of youth aged 15–24 years in Nepal [30]. Furthermore, 98.3% of the participants had at least primary-level education, indicating that the participants were capable of understanding health-related information. However, communicative literacy and critical literacy scale scores were low, suggesting that low health literacy may be related to one of the factors contributing to health issues remaining unresolved and becoming more complex despite the growing educational system among the younger generation in Nepal.

Information is a necessary grounding material for personal decision-making. Autonomous decision-making, including obtaining reliable information, selecting the appropriate information for one's own situation from a list of options, and making autonomous decisions, is the desired health behavior. However, Nepalese women's autonomy in household decision-making is reported to be low in all aspects of managing their own health, making major household purchases, purchasing daily necessities, and visiting family and relatives [31].

In other words, if health literacy can be strengthened with effective methodologies to empower Nepalese women to take interest in their own health, seek health information on their own, discern and select information, communicate it to others, and adapt it to their own lives, they will significantly contribute to maintaining and improving not only their health but also the health of their families.

Interventions for the younger generation are particularly useful, and health behaviors are more likely to be sustained; thus, it is hoped that interventions to improve health literacy will become widespread as preconception care [32]. If Nepalese women can improve their health, including improving nutrition from preconception, they can prevent perinatal complications,

improve birth outcomes, and maintain and promote fetal health. Preconception care, which includes improving nutrition from adolescence, delaying the time of conception, and optimizing pregnancy spacing, is important in Nepal because majority of the pregnant women are teenage mothers or in their early 20s. Antenatal care (ANC) also ensures continuity in maternal and fetal health care, but no health education for individuals or groups of pregnant women is provided. Continuous health education in ANC is important to further improve maternal health in Nepal.

This study found that art-based material and face-to-face tutoring improved the health literacy of Nepalese pregnant women and that improved health literacy contributed to improved anemia among pregnant women. Health literacy was found to be an important factor in improving the nutritional status of Nepalese pregnant women.

Currently, the nutritional challenges of Nepalese pregnant women are becoming more complex. Comprehensive nutritional assessment using multiple indicators is needed to determine the extent and characteristics of nutritional disorders. This indicates an urgent need for nutrition assessment and for review of existing nutritional support to all generations in developing countries. This study focused on health literacy as well as literacy and an intervention. The study revealed the importance of health literacy as a determinant of the nutritional status of anemic Nepalese pregnant women and showed the effectiveness of the intervention. Health literacy was identified as an important factor in supporting nutrition improvement for Nepalese pregnant women even though literacy rates have increased. Furthermore, the study showed the importance of evidence-based knowledge dissemination using effective educational material. The material employed in this study was designed using pictures and diagrams that are easy for pregnant women and families with low health literacy to understand visually.

## Limitations

In this study, only a self-reporting procedure was used to confirm iron medication behavior, which limited tracking. There is a need to implement a research procedure to count iron tablet sheets in the future to accurately confirm medication adherence. In addition, this was a single-center RCT study, and future long-term multicenter observational studies are desirable.

Furthermore, to determine the cause of anemia, not only the Hb level but also a comprehensive evaluation of other parameters, including hematocrit, mean corpuscular volume, red blood cell, mean corpuscular Hb, mean corpuscular Hb concentration, red cell distribution width, and ferritin, is necessary.

## Conclusions

The results of this study showed that individualized health education using pictures, photographs, and nomograms without text was effective in improving Hb levels among pregnant Nepalese women. A Nepali version of the HLS-14 was developed as a literacy measurement tool and found to be useful for evaluating health literacy levels before and after the intervention. The women who received education sessions showed significantly higher improvement in overall health, functional, communicative, and critical literacy scores than the distribution and control groups.

## Supporting information

**S1 Checklist. CONSORT 2010 checklist of information to include when reporting a randomised trial\*.**
(DOC)

**S1 File.**
(DOCX)

**S2 File. Inclusivity in global research.**
(DOCX)

## Acknowledgments

We would like to express our deepest gratitude to the Nepalese pregnant women who participated in the survey. We also extend our utmost respect and appreciation for the active participation of the intervention study participants and for the cooperation and support of the five research nurses.

## Author Contributions

**Conceptualization:** Hiroko Sakai.

**Data curation:** Rina Kawata.

**Formal analysis:** Hiroko Sakai, Rina Kawata.

**Funding acquisition:** Hiroko Sakai.

**Investigation:** Hiroko Sakai, Rajesh Adhikari, Yoko Oda Thapa.

**Methodology:** Hiroko Sakai, Rajesh Adhikari, Yoko Oda Thapa.

**Project administration:** Hiroko Sakai, Rajesh Adhikari, Yoko Oda Thapa, Tulsi Ram Bhandari.

**Software:** Hiroko Sakai.

**Supervision:** Tulsi Ram Bhandari.

**Validation:** Hiroko Sakai.

**Visualization:** Hiroko Sakai.

**Writing – original draft:** Hiroko Sakai, Rina Kawata, Yoko Oda Thapa.

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
