## [Decision Letter · Decision Letter 0]

6 Oct 2023

PONE-D-22-35514Effectiveness of art-based health education on anemia and health literacy among pregnant women in Western Nepal: A randomized controlled trialPLOS ONE

Dear Dr. Sakai,

Thank you for submitting your manuscript to PLOS ONE. After careful consideration, we feel that it has merit but does not fully meet PLOS ONE’s publication criteria as it currently stands. Therefore, we invite you to submit a revised version of the manuscript that addresses the points raised during the review process.

The manuscript has been evaluated by two reviewers, and their comments are available below.

The reviewers have raised a number of major concerns. They feel the manuscript should outline a clearly-defined research question, and they request improvements to the reporting of methodological aspects of the study, for example, regarding the exclusion criteria and more information on how the data collection was completed. The reviewers also note concerns about the statistical analyses presented and request re-analyses be completed.

Could you please carefully revise the manuscript to address all comments raised?

In addition, please clarify the following-

1. Clinical trial registration appears to have been completed retroactively and should be clarified in Line105.2. Ethics Committee approval is in  both Japanese and English and it is difficult to review the CONSORT design (L120).  Please provide additional copy in English.  3. In Data Collection/Sample Size, a previous study is mentioned (L276) and should be cited.4. In Results and Table 1,  the description of caste (L315-317) should match the description in Table 1.5.  In Table 1, "Occupation" (which addresses both husband and wife) could be moved next to "Occupation" for clarity ==============================

We look forward to receiving your revised manuscript.

Kind regards,

Avanti Dey

Staff Editor

PLOS ONE

Journal Requirements:

3. Thank you for submitting your clinical trial to PLOS ONE and for providing the name of the registry and the registration number. The information in the registry entry suggests that your trial was registered after patient recruitment began. PLOS ONE strongly encourages authors to register all trials before recruiting the first participant in a study.

1) your reasons for your delay in registering this study (after enrolment of participants started);

2) confirmation that all related trials are registered by stating: “The authors confirm that all ongoing and related trials for this drug/intervention are registered”.

4. We noted in your submission details that a portion of your manuscript may have been presented or published elsewhere. [Yes, we are planning to write a report for the official publication of the Health Education Association of Nepal (HEAN) using Table 2.] Please clarify whether this [conference proceeding or publication] was peer-reviewed and formally published. If this work was previously peer-reviewed and published, in the cover letter please provide the reason that this work does not constitute dual publication and should be included in the current manuscript.

“This work was supported by JSPS KAKENHI Grant Numbers 18k17592,22k0026.”

“•           Initials of the authors who received each award: H S

•             Grant numbers awarded to each author: Hiroko Sakai JSPS KAKENHI Grant Numbers 18k17592,22k0026.

•             The full name of each funder: Grant-in-Aid for Young Scientists, Fostering Joint International Research (B)

•             URL of each funder website: Japan Society for the Promotion of Science (jsps.go.jp)

Additional Editor Comments (if provided):

This research regarding health education on anemia and health literacy among pregnant women in Nepal has merit but needs minor clarifications, i.e.,

1. Clinical Trial registration appears to have been completed retroactively, and should be clarified in Line 105 (L1050.

2. Ethics Committee approval is in both Japanese and English, and it is difficult to review the CONSORT design (L120).

3. In Data Collection/Sample Size, a previous study is mentioned (L276) and needs to be cited.

4. In Results and Table 1, the description of caste (L315-317) should match the description in Table 1(L322).

5. In Table 1, "Occupation" (which addresses both husband and wife) could be moved next to "Income" to prevent confusion.

Reviewers' comments:

Reviewer's Responses to Questions

**Comments to the Author**

1. Is the manuscript technically sound, and do the data support the conclusions?

Reviewer #1: Yes

Reviewer #2: Yes

2. Has the statistical analysis been performed appropriately and rigorously? 

Reviewer #1: Yes

Reviewer #2: Yes

3. Have the authors made all data underlying the findings in their manuscript fully available?

Reviewer #1: No

Reviewer #2: Yes

4. Is the manuscript presented in an intelligible fashion and written in standard English?

Reviewer #1: Yes

Reviewer #2: Yes

5. Review Comments to the Author

Reviewer #1: A three-arm randomized controlled clinical trial was conducted which aimed to assess the effectiveness of a face-to-face educational intervention in reducing anemia in pregnant Nepalese women. When comparing mean change in Hb levels from pre to post intervention the education group exhibited statistically significantly higher levels.

Minor revisions:

1- Abstract: Typographic error: Remove the phrase “in the group” which follows “0.96 +/- 1.21.”

2- Line 290: For clarity consider stating, “Continuous variables were summarized using mean and standard deviation (SD)."

3- Line 292: Grammatical error: Replace “between” with “among.”

4- Line 297: To conform to convention, replace “probability values” with “p-values.”

5- Line 145 states that the results were analyzed using Microsoft Excel, and line 299 states that data were analyzed using SPSS. Please clarify.

6- The standard statistical term for “average” is “mean.”

7- The p-value of 0.035 from the ANOVA test indicates statistically different means among the three groups. Indicate the statistical method used for making pairwise comparisons to determine which pairs differed.

8- Table 4 results: State the statistical method and provide results for analyzing these four outcomes when comparing the three randomization arms.

Reviewer #2: This finding add significants information to the literature of anemia prevention interventions

Some improvements is needed in the introductions, method, results and discussions, please find the detail comment in the file provided

6. PLOS authors have the option to publish the peer review history of their article (what does this mean?). If published, this will include your full peer review and any attached files.

Reviewer #1: No

Reviewer #2: No

---

## [Author Response · Author response to Decision Letter 0]

9 Feb 2024

Response to reviewers has been uploaded separately

---

## [Decision Letter · Decision Letter 1]

13 Mar 2024

PONE-D-22-35514R1Effectiveness of art-based health education on anemia and health literacy among pregnant women in Western Nepal: A randomized controlled trialPLOS ONE

Dear Dr. Sakai,

Thank you for submitting your manuscript to PLOS ONE. After careful consideration, we feel that it has merit but does not fully meet PLOS ONE’s publication criteria as it currently stands. Therefore, we invite you to submit a revised version of the manuscript that addresses the points raised during the review process.

We look forward to receiving your revised manuscript.

Kind regards,

Thomas Kwasi Gyan

Academic Editor

PLOS ONE

Journal Requirements:

Reviewers' comments:

Reviewer's Responses to Questions

**Comments to the Author**

1. If the authors have adequately addressed your comments raised in a previous round of review and you feel that this manuscript is now acceptable for publication, you may indicate that here to bypass the “Comments to the Author” section, enter your conflict of interest statement in the “Confidential to Editor” section, and submit your "Accept" recommendation.

Reviewer #1: (No Response)

Reviewer #2: (No Response)

2. Is the manuscript technically sound, and do the data support the conclusions?

Reviewer #1: (No Response)

Reviewer #2: (No Response)

3. Has the statistical analysis been performed appropriately and rigorously? 

Reviewer #1: (No Response)

Reviewer #2: (No Response)

4. Have the authors made all data underlying the findings in their manuscript fully available?

Reviewer #1: (No Response)

Reviewer #2: (No Response)

5. Is the manuscript presented in an intelligible fashion and written in standard English?

Reviewer #1: (No Response)

Reviewer #2: (No Response)

6. Review Comments to the Author

Reviewer #1: (No Response)

Reviewer #2: (No Response)

7. PLOS authors have the option to publish the peer review history of their article (what does this mean?). If published, this will include your full peer review and any attached files.

Reviewer #1: No

Reviewer #2: **Yes: **Izzatul Arifah

---

## [Author Response · Author response to Decision Letter 1]

4 Apr 2024

Response to Reviewers

Journal Requirements: Please review your reference list to ensure that it is complete and correct. If you have cited papers that have been retracted, please include the rationale for doing so in the manuscript text, or remove these references and replace them with relevant current references. Any changes to the reference list should be mentioned in the rebuttal letter that accompanies your revised manuscript. If you need to cite a retracted article, indicate the article’s retracted status in the References list and also include a citation and full reference for the retraction notice.

We have checked the reference list and ensured that it is complete and correct. We confirm that no retracted papers have been cited.

---

## [Editor Report · Decision Letter 2]

19 Jun 2024

PONE-D-22-35514R2Effectiveness of art-based health education on anemia and health literacy among pregnant women in Western Nepal: A randomized controlled trialPLOS ONE

Dear Dr. Sakai,

Thank you for submitting your manuscript to PLOS ONE. After careful consideration, we feel that it has merit but does not fully meet PLOS ONE’s publication criteria as it currently stands. Therefore, we invite you to submit a revised version of the manuscript that addresses the points raised during the review process.

Please submit your revised manuscript by Aug 03 2024 11:59PM. If you will need more time than this to complete your revisions, please reply to this message or contact the journal office at plosone@plos.org. Please include the following items when submitting your revised manuscript:A rebuttal letter that responds to each point raised by the academic editor and reviewer(s). You should upload this letter as a separate file labeled 'Response to Reviewers'.A marked-up copy of your manuscript that highlights changes made to the original version. You should upload this as a separate file labeled 'Revised Manuscript with Track Changes'.An unmarked version of your revised paper without tracked changes. You should upload this as a separate file labeled 'Manuscript'.We look forward to receiving your revised manuscript.

Kind regards,

Thomas Kwasi Gyan

Academic Editor

PLOS ONE

Journal Requirements:

**Additional Editor Comments:**

A few clarifications required. Other sections also required a bit of strengthening including the introductions, method, results and discussions as follows:

Abstract: Typographic error: Remove the phrase “in the group” which follows “0.96 +/- 1.21.”Line 290: For clarity consider stating, “Continuous variables were summarized using mean and standard deviation (SD)."Line 292: Grammatical error: Replace “between” with “among.”Line 297: To conform to convention, replace “probability values” with “p-values.”145 states that the results were analyzed using Microsoft Excel, and line 299 states that data were analyzed using SPSS. Please clarify.The standard statistical term for “average” is “mean.”The p-value of 0.035 from the ANOVA test indicates statistically different means among the three groups. Indicate the statistical method used for making pairwise comparisons to determine which pairs differed.Table 4 results: State the statistical method and provide results for analyzing these four outcomes when comparing the three randomization arms.Line 85. Please add a prove (previous study) showing that high literacy level affect maternal anemiaLine 87: How prevalence low health literacy in Nepalese people or Nepalese women in particular or pregnant Nepalese women. Please add data to prove statementLine 109: Add clinical trial registration numberLine 122: Please add information on who was blinded in the study is it the participants or investigatorLine 193: How long was each education session given? And how much frequency in total during the intervention period.Line 307: Please add information how many respondents per groupTable 1: Results show comparability between intervention and control groupTable 3: Add the post hoc test results for the three groupsTable 4: Add the statistical test score at baseline and endlineLine 420: What is the implication of the study findings to the implementation of ANC delivery or policy on the ANC delivery

---

## [Author Response · Author response to Decision Letter 2]

15 Aug 2024

we had previously submitted the paper as requested in the journal email

---

## [Editor Report · Decision Letter 3]

16 Sep 2024

Effectiveness of art-based health education on anemia and health literacy among pregnant women in Western Nepal: A randomized controlled trial

PONE-D-22-35514R3

Dear Dr. Sakai,

We’re pleased to inform you that your manuscript has been judged scientifically suitable for publication and will be formally accepted for publication once it meets all outstanding technical requirements.

Kind regards,

Thomas Kwasi Gyan

Academic Editor

PLOS ONE
---

## [Editor Report · Acceptance letter]

20 Sep 2024

PONE-D-22-35514R3 

PLOS ONE

Dear Dr. Sakai, 

I'm pleased to inform you that your manuscript has been deemed suitable for publication in PLOS ONE. Congratulations! Your manuscript is now being handed over to our production team.

Kind regards, 

on behalf of

Dr. Thomas Kwasi Gyan 

Academic Editor

PLOS ONE